# Effect of Various Defects on 4H-SiC Schottky Diode Performance and Its Relation to Epitaxial Growth Conditions

**DOI:** 10.3390/mi11060609

**Published:** 2020-06-24

**Authors:** Jinlan Li, Chenxu Meng, Le Yu, Yun Li, Feng Yan, Ping Han, Xiaoli Ji

**Affiliations:** 1College of Information Engineering, Yangzhou University, Yangzhou 225009, China; 007328@yzu.edu.cn; 2College of Electronic Science and Engineering, Nanjing University, Nanjing 210093, China; mg1823040@smail.nju.edu.cn (C.M.); yule@nju.edu.cn (L.Y.); fyan@nju.edu.cn (F.Y.); 3Science and Technology on Monolithic Integrated Circuits and Modules Laboratory, Electronic Devices Institute, Nanjing 210016, China; yuefei_2@126.com

**Keywords:** Ni/4H-SiC Schottky barrier diodes (SBDs), C/Si ratios, 1/*f* noise

## Abstract

In this paper, the chemical vapor deposition (CVD) processing for 4H-SiC epilayer is investigated with particular emphasis on the defects and the noise properties. It is experimentally found that the process parameters of C/Si ratio strongly affect the surface roughness of epilayers and the density of triangular defects (TDs), while no direct correlation between the C/Si ratio and the deep level defect Z_1/2_ could be confirmed. By adjusting the C/Si ratio, a decrease of several orders of magnitudes in the noise level for the 4H-SiC Schottky barrier diodes (SBDs) could be achieved attributing to the improved epilayer quality with low TD density and low surface roughness. The work should provide a helpful clue for further improving the device performance of both the 4H-SiC SBDs and the Schottky barrier ultraviolet photodetectors fabricated on commercial 4H-SiC wafers.

## 1. Introduction

As an excellent wide bandgap semiconductor material, 4H-SiC has attracted continuous attention in the last several decades. Due to its wide bandgap, high thermal conductivity, high saturated electron drift velocity and high physical and chemical stability, 4H-SiC is an ideal material for high-performance ultraviolet (UV) photodetectors available in high-temperature environments, as well as power devices for high-temperature and high-frequency applications [1,2,3,4,5,6]. The device performance of both the photodetectors and the power devices greatly depends on the epitaxial quality of commercial 4H-SiC wafers, which has been significantly improved in recent years. However, due to substrate material imperfection and epitaxial growth immaturity, there are still various defects in the large-size 4H-SiC epilayers, such as morphological defects (micropipes, downfalls, particles, triangular defects (TDs), carrots, etc.), crystallographic defects (stacking faults, threading dislocations, etc.) and deep level defects, etc. [7,8,9,10,11,12,13,14,15], which will inevitably have a serious impact on the performance of 4H-SiC power devices and ultraviolet photodetectors. Among the morphological defects, micropipes have serious impact on the device performance but are preventable, since the substrates with micropipe density less than 0.1 cm^−2^ are commercially available [16,17,18]. In our previous paper, we reported the density of other morphological defects in a 4-inch 4H-SiC epitaxial wafer, as well as their probability of causing Schottky diode power device failure [19]. The average morphological defect (including TDs, downfalls, particles, carrots) density in the 4-inch 4H-SiC epitaxial layers is 1.25 cm^−2^, in which the TDs, the downfalls, the particles, and the carrots account for approximately 60%, 2%, 31% and 7%, respectively. In addition, the probability of causing device failure is 100% for TDs and downfalls, while the probability of degrading the reverse breakdown voltage of the device is 2% and 30% for particles and carrots, respectively. Therefore, among all these morphological defects, the TDs should attract enough attention as the critical device killer. Meanwhile, the deep level defect Z_1/2_ center (located at 0.6~0.7 eV below the conduction band of 4H-SiC) is now considered to be the main life killer of n-type 4H-SiC devices. The concentration of the Z_1/2_ is (0.1–4) × 10^13^ cm^−3^ for the commercially available 4H-SiC material, which seriously affects the performance of bipolar devices and photodetectors due to the existence of deep level traps [20]. Thus, the TDs and the Z_1/2_ defects which widely exist in commercially available 4H-SiC epilayers are usually considered as two significant factors that hinder the device performance. The structures of TDs are related to the presence of foreign particle inclusion at the epi-substrate interface, resulting in an increase of leakage current and the reduction of breakdown voltage in the SiC power devices [21]. The Z_1/2_ defect at E_c_-0.6 eV originates from carbon vacancies, which may affect the barrier height and leakage current of SiC Schottky barrier diodes (SBDs) [22]. On the other hand, all these defects may increase the recombination of electron-hole pairs and enhance the surface recombination in the Schottky barrier UV photodetectors, leading to a lower responsivity of the device [23,24]. A large number of studies have shown that the formation of these defects is closely associated with growth temperature, C/Si ratio, thermal oxidation, carbon ion implantation, thermal annealing, in-situ pre-growth etch and cutting angle of SiC substrate, etc. [25,26,27,28,29,30].

Therefore, the optimization of growth conditions is critical to the reduction of defects and the improvement of the device performance. Miyazawa reported that the reduction of the Z_1/2_ defect concentration is achieved mainly through thermal oxidation/thermal annealing or C^+^-implantation/thermal annealing, unfortunately, new traps are introduced by these two methods [31]. Yazdanfar pointed out that TD density can be effectively reduced by in-situ pre-growth etch before growth at lower C/Si ratio and growth temperature [32]. However, these methods are very complicated and impractical for industrial processes. In practicality, the variations of growth conditions influence multiple defects in the 4H-SiC epilayers, and consequently determine the device performance in an intricate way. It is necessary to survey how these parameters (e.g., C/Si ratio, growth rate and doping density) affect TDs and Z_1/2_ defects, which are the most commonly exsiting defects in the commercial 4H-SiC wafers. Meanwhile, the key issue of process parameters optimization is to seek out the critical factors affecting device performance; therefore, it is also important to establish a direct link between the growth parameters and the device performance. By analyzing the results of the electrical parameters of the device performance, not only can the most important defect information affecting the performance be identified, but it can also provide effective suggestions for further optimization of the growth parameters.

In this paper, we investigated the influence of different chemical vapor deposition (CVD) growth conditions on the defect density of 4H-SiC epilayers and the electrical properties of Ni/4H-SiC SBDs, with particular attention to the TDs and the Z_1/2_ defects. The integration of scanning electron microscopy (SEM), atomic force microscopy (AFM), micro-photoluminescence (PL), and micro-Raman was used to study the mechanism of defect evolution in 4-inch 4H-SiC epilayers. The combination of reverse leakage current testing, noise measurement, and the deep level transient spectrum testing (DLTS) was used to reveal the root cause of the growth conditions impacting on device performance.

## 2. Materials and Methods

4H-SiC epilayers with different C/Si ratios (C/Si = 0.9, 1 or 1.1) were homoepitaxially grown on commercially available 4H-SiC substrates in a SiH_4_-C_3_H_8_-H_2_ chemical vapor deposition (CVD) system (AIXTRON VP2400, AIXTRON, Herzogenrath, Germany). In this study, all substrates were commercial chemo-mechanically polished micropipe-free 4-inch 4H-SiC. The substrate is highly doped with nitrogen and cut off at 8° towards [11$\bar{2}$0] direction. Here, SiH_4_, C_3_H_8_ and HCl were used as source gases while H_2_ was selected for dilution and carrier gas. The C/Si ratio was varied from 0.9 to 1.1 by changing the C_3_H_8_ flow rate at a fixed SiH_4_ flow rate. The typical epitaxial growth rates were fixed at 60 μm/h, and the epitaxial temperature and growth pressure were controlled within 1550~1600 °C and 50~150 mbar, respectively. Nitrogen gas was used as an n-type dopant, and the typical doping concentration was 1 × 10^15^ cm^−3^. Additional 4-inch 4H-SiC epilayers with different growth rate (*v* = 30 μm/h and 60 μm/h) and doping concentration (*N_d_* = 4 × 10^15^ cm^−3^, 7.5 × 10^15^ cm^−3^ and 1 × 10^16^ cm^−3^) were also prepared for exploring the dependence of the process parameters on deep defect density, respectively. All the samples (#1~#7) have the same thickness of 12 μm, with different growth parameters summarized in Table 1. After the homoepitaxial growth in CVD system, a 100 nm Ni Ohmic contact was formed by annealing in nitrogen at 1000 °C for 5 min in the back sides of the device, and a 75 nm Ni Schottky contact layer was sputtered with an active area of 1 mm^2^ in the front sides of the device.

The room temperature micro-PL and micro-Raman spectroscopy measurement were performed on the samples excited by a 325 nm He-Cd laser, where the laser beam was focused to a spot of 10 μm diameter using a sapphire objective lens. Moreover, the atomic force microscopy (AFM) (WET-SPM-model, SPM-9600, Shimadzu Corp., Kyoto, Japan) imaging with a metalized cantilever was carried out for characterizing the surface roughness of wafers. The *C*-*V* characteristic measurements (Keithley 4200, Keithley, Cleveland, OH, USA) were conducted at a frequency of 1 MHz to determine the effective doping density of devices based on 1/*C^2^* vs. *V* plots. The noise spectrum was tested to study device noise performance. Deep level characterization of the 4H-SiC SBDs were carried out using DLS-83D Deep Level Transient Spectroscopy test system (Semilab, Budapest, Hungary). Temperature ranging from 77 K to 550 K was selected by a ACP-4000 temperature controller at a heating rate of 0.1 K/s.

## 3. Results and Discussion

Figure 1a shows the room temperature micro-PL spectra corresponding to the A and B positions in the illustration for the 4H-SiC epilayer grown with C/Si = 1.1. The inset SEM image at position A exhibits the crystal structure region of the 4H-SiC epilayer without TDs. The strongest peak of 391 nm corresponding to 3.17 eV is attributed to the typical band edge emission of 4H-SiC [33]. However, at the apex of the TDs (position B), the 4H-SiC band edge emission intensity at 391 nm was significantly reduced. Furthermore, an additional small emission peak appeared at 423 nm corresponding to 2.93 eV, which is consistent with the emission wavelength of the stacking fault in the 4H-SiC epilayer. This stacking fault is determined by the different stacking order, which is defined as a single Shockley SF (1SSF) using the Zhdanov notation [34,35]. Figure 1b shows typical micro-Raman spectra of 4H-SiC at points A and B. The results show two FTO modes at 776 cm^−1^ and 796 cm^−1^, and an FLO mode at 964 cm^−1^, which is consistent with the typical Raman spectra of 4H-SiC [36]. However, for the TD region (point B), the intensity of the Raman peak at 796 cm^−1^ is gradually increased. Since the 3C-SiC also has a TO peak at 796 cm^−1^, the presence of the 3C-SiC polytype could be inferred. The ratio of the two peaks can be taken as a measure of the presence of the 3C-SiC polytype [37]. Therefore, we can confirm that stacking faults and the 3C-SiC are the reasons of nucleation at the triangle defect vertex. Figure 2 shows the dependence of TD density on the C/Si ratios for 4H-SiC epilayers. The wafers (#1~#3) were inspected using optical microscopy to evaluate the density of TDs. The results show that the density of TDs is reduced from 1.3 cm^−2^ to 0.13 cm^−2^ with the C/Si ratio decreasing from 1.1 to 0.9. Kojima et al. have reported that the origin of the TDs is attributed to step-bunching caused by the interrupted step-flow growth [38]. In the carbon-rich growth environment (C/Si = 1.1), the surface free energy of the crystal surface is higher than that under silicon-rich conditions, so it is necessary to reduce the surface free energy by the formation of step-bunching [39]. Thus, there is a higher probability of two-dimensional nucleation due to the suppression of the step-flow growth at high C/Si ratios. Consequently, a relatively low C/Si ratio can effectively reduce the TD density.

It is well known that TDs are fatal to power devices, directly causing device failure. Moreover, although the TDs could be got rid of in most areas of the epitaxial wafer by a proper C/Si ratio, there still exist some hidden defects that affect device performance, such as interface states and deep defects, etc. Therefore, we selected the devices without TDs on the epilayer region to further investigate the effect of hidden defects on the performance of 4H-SiC SBDs prepared on epilayers with different C/Si ratios (C/Si = 0.9, 1 or 1.1). Figure 3a shows the mean values of the reverse current density varying with the C/Si ratios under the reverse bias voltage of −200 V. It is noted that there is still a minor difference in device performance due to process unevenness even in the same wafer. In order to eliminate the influence of the performance non-uniformity between devices, we analyzed the average reverse *I*-*V* results of five samples randomly selected on each wafer, and Ni/4H-SiC SBDs with C/Si = 1 is shown as an example in the inset of Figure 3a. It is found that the average values of the reverse current density of Ni/4H-SiC SBDs with C/Si = 0.9, 1 or 1.1 are 1.8 × 10^−12^ A/cm^2^, 5.6 × 10^−12^ A/cm^2^, and 1.5 × 10^−11^ A/cm^2^, respectively. With the increasing of C/Si ratios, the reverse leakage current gradually increases. Compared with the sample with C/Si = 0.9, the leakage current of the sample with C/Si = 1.1 increases by nearly an order of magnitude. Figure 3b shows the forward *I*-*V* characteristics of Ni/4H-SiC SBDs under different C/Si ratios. It is seen that all samples have relatively uniform and consistent forward *I*-*V* curves. According to the thermionic emission (TE) theory, the relationship between current and voltage is defined by [40]
(1)$${J=A}^{*}T^{2}\exp(\frac{{-q\Phi }_{B}}{\mathrm{kT}})\;[\exp\frac{\mathrm{qV}}{\mathrm{nkT}}-1]$$

Here, the values of the barrier height *Φ_B_* and the ideality factor *n* can be calculated by
(2)$$n=\frac{q}{\mathrm{kT}}(\frac{\mathrm{dV}}{d\ln J})$$
(3)$$\Phi_{B}=\frac{\mathrm{kT}}{q}\ln(\frac{A^{*}T^{2}}{J_{s}})$$
where *n* is the ideality factor, *V* is the applied voltage, *Φ_B_* is the zero bias Schottky barrier height and *A** is the effective Richardson constant with 146 A/cm^2^K^2^ for 4H-SiC. The values of ideality factor *n* and barrier height *Φ_B_* obtained from *I*-*V* characteristics, and the values of effective doping density *N_eff_* extracted from *C*-*V* characteristics in the inset of Figure 3b were summarized in Table 2. The *n* and *Φ_B_* are all approximately 1 and 1.63 eV, showing good rectification characteristics. Moreover, the *N_eff_* are in the range of 1.15~1.25 × 10^15^ cm^−3^. Therefore, the barrier height and the effective doping density are not the root cause of the increase in reverse leakage current as the C/Si ratios increases, because of the same barrier height for all samples. As is known, the leakage current of devices may be affected by a lot of factors, including surface defects, crystal defects, deep level defects, interface states, and crystal quality, etc. [41,42,43]. Combined with the above analysis, we can infer that the epitaxy process with different C/Si ratios may affect the interface state, deep level defects or crystal quality of the epilayer, resulting in different reverse leakage current changing with C/Si ratios.

Figure 3c show the frequency and bias voltages dependence of the spectral noise density (S_I_) for the Ni/4H-SiC SBDs with C/Si=0.9 at room temperature under various reverse bias voltages (*V_R_* = −10~(−200) V). It is clearly observed that the spectral noise density is inversely proportional to the frequency at different voltages and increases approximately linearly with the increase of the reverse bias voltage. It has the form of 1/*f*^α^ noise, where α=1, its characteristics is flicker noise (1/*f* noise). This dependence is consistent with the reported results of SiC Schottky diodes (*α* = 0.5~1.5) [44,45]. The main contribution to 1/*f* noise and resistance noise comes from the Schottky barrier. Figure 3d shows the noise spectra of the Ni/4H-SiC SBDs with varying C/Si ratios (C/Si = 0.9, 1 or 1.1) under *V_R_* = −150 V. It displays 1/*f* behavior at frequencies below 1K Hz for all samples. Furthermore, the spectral noise density increases with the increase of the C/Si ratio, and the minimum is obtained at C/Si = 0.9. However, for the samples with C/Si = 1 and 1.1, the noise spectra become almost frequency independent at frequencies greater than 1K Hz. Several orders of magnitude increase in the noise are caused by superimposed thermal noise (S_I_ = 4*K_0_Tg*) due to larger reverse current in samples with C/Si = 1 or 1.1, which is in agreement with the results of Figure 3a. The 1/*f* noise is mainly affected by the factors such as the interface state and crystal quality of the semiconductor material. Zhang et al. reported that the 1/*f* noise is reduced significantly with the increase of temperature in the SiC MOSFET within a temperature range of 85~510 K, which is attributed to the interface trap density decreasing with the temperature [46]. Soibel et al. also revealed that the 1/*f* noise in the InAs/GaSb superlattice detector is related to the side leakage current caused by the surface states. The larger side leakage current is induced by the higher surface state density, leading to a significant increase in noise [47]. Accordingly, our results show that as the C/Si ratio decreases, the spectral noise density decreases by about three orders of magnitude (C/Si = 0.9). The influence of C/Si ratios on device noise should be due to the surface state density or the interface state density caused by different crystal quality of 4H-SiC epilayers. Therefore, an optimized C/Si ratio not only effectively improves the performance of SBDs, but should also improves the responsivity of the Schottky barrier UV photodetector due to the reduction of noise.

On the other hand, the Z_1/2_ is the most important deep level defect in the grown N-type 4H-SiC epilayer. The presence of deep level defects acts as charge trapping, capturing photogenerated carriers or increasing their recombination probability in the depletion region of the Schottky barrier UV photodetectors, leading to a significant reduction of responsivity [48]. Moreover, the trap or defect-assisted tunneling can result in the increase of leakage current and the reduction of carrier lifetime. To further understand the effects of deep defects on the electrical properties of the Ni/4H-SiC SBDs with different growth parameters, DLTS measurements were carried out in a temperature range of 80~550 K. Figure 4a shows the representative DLTS spectra of the 4H-SiC SBD grown with a C/Si ratio of 0.9, under the reverse bias *V_R_* = −3 V, pulse voltage *V_p_* = 0 V, pulse width *t_p_* = 50 μs and frequency *f* = 80~960 Hz. It can be seen that the only peak appears in the DLTS spectrum, which can be attributed to Z_1/2_ as an intrinsic defect in the SiC epilayer. The microstructure of Z_1/2_ has been extensively studied, such as carbon vacancies (V_c_), silicon vacancies (V_si_), reverse (C_si_, Si_c_) or more likely defect complexes (C_si_ + Si_c_, V_c_ + V_si_) [49]. Eberlein et al. have reported the Z_1/2_ is composed of the traps Z_1_ and Z_2_, and such centers reveal the negative-U character with donor (0/+) levels located at E_c_-0.43(0.46) eV and acceptor (−/0) levels situated at E_c_-0.67(0.71) eV [50]. The Z_1/2_ has recently been showned to be most likely an acceptor level of V_c_ [51].Besides, the variation of Z_1/2_ defect concentration in all samples (#1~#7) is shown in Figure 4b. The parameters of defects are summarized in Table 3. The results show that with the change of C/Si ratio (C/Si = 0.9~1.1), the Z_1/2_ defect concentration is 2.28 × 10^13^ cm^−3^, 3.79 × 10^13^ cm^−3^, and 2.22 × 10^13^ cm^−3^, respectively, indicating that the Z_1/2_ defect concentration remains substantially unchanged. Litton et al. reported that the Z_1/2_ concentration decreased with increasing carbon to silicon ratios of one, three and six due to the reduction of C vacancies under a C-rich growth condition. However, as the carbon-to-silicon ratio increases, the TD density of its epitaxial wafers will increase sharply, which will directly cause the decrease of device performance. Compared to our experimental results, the Z_1/2_ defect density does not change significantly with the change of the carbon-to-silicon ratio may be due to the small range of change of the carbon-to-silicon ratio [49]. In addition, as the doping concentration and growth rate increase, the Z_1/2_ defect concentration ranges between 6.78 × 10^12^ cm^−3^ and 1.41 × 10^13^ cm^−3^. Lilja et al. reported that the Z_1/2_ concentration does not show any obvious tendency with epitaxial growth rate, and the concentration of Z_1/2_ in different epitaxial wafers is between 1 × 10^13^ cm^−3^ and 4 × 10^13^ cm^−3^ [52]. The above literature results is in agreement with our results. Therefore, the above experimental results show that the concentration of Z_1/2_ intrinsic defect does not change significantly with the process parameters. The inset of Figure 4b shows the dependence of the reverse current density (under *V_R_* = −200 V) and breakdown voltage as a function of Z_1/2_ defect density for Ni/4H-SiC SBDs. It can be seen from the figure that the reverse current and the breakdown voltage have no obvious trend with the increase of Z_1/2_ defect density. However, as the carbon-to-silicon ratio decreases from 1.1 to 0.9, the reverse current gradually decreases, and the breakdown voltage slightly increases, which may be due to the optimization of interface quality leading to improved device performance. In addition, devices with different C/Si ratios exhibit uniform and good rectification characteristics. (the *n* and *Φ_B_* are all approximately 1 and 1.63 eV, respectively). Therefore, the Z_1/2_ defect is not the root cause of device performance degradation.

In order to further study the effect of C/Si ratio on the reverse leakage and noise of the device, Figure 5a shows the AFM images of the 4H-SiC epilayers grown with different C/Si ratios from 1.1 to 0.9. It is exhibited that the strong dependence of surface morphology of 4H-SiC epilayers on the C/Si ratios. It is observed that the surface morphology is rather smooth with gentle undulation under Si-rich growth environment (C/Si = 0.9). The root-mean-square (RMS) roughness is 0.8 nm at a C/Si ratio of 1.1, and it is decreased to 0.15 nm at C/Si ratio of 0.9 in a 3 μm × 3 μm scan area. As seen in Figure 5b, the PL spectra of 4H-SiC epilayers with different C/Si ratios shows that the 4H-SiC band edge emission intensity (391 nm) of the sample with C/Si = 1.1 is significantly weaker than that of the sample with C/Si = 0.9, which is due to the fact that the non-radiative recombination caused by the defects severely weakens the band edge emission intensity of 4H-SiC [34]. Hence, we can see that the sample with C/Si = 0.9 has minimal roughness and defects, resulting in minimum noise. By comparison, in the samples of C/Si = 1 or 1.1, the quality of the wafer becomes worse, thus causing more noise.

Combined with the above test result, we can see that the roughness of the epitaxial wafer and the crystal quality associated with the defects play an important role in device performance. Since the Z_1/2_ defect density and barrier height of 4H-SiC SBDs do not change with the different CVD growth conditions, the most probable reason for the difference in reverse leakage and noise of the device should be the different interface state densities caused by the different epitaxial quality. Therefore, the optimization of the C/Si ratio not only effectively reduces the TD density but also improves the performance of 4H-SiC SBD.

## 4. Conclusions

The impacts of CVD growth parameters on defect density of 4-inch 4H-SiC epilayers and performance of Ni/4H-SiC SBDs were studied. It is found that the reverse current and noise characteristics of 4H-SiC SBDs are highly dependent on the C/Si ratio. As the C/Si ratio grows from 0.9 to 1.1, the average reverse current and 1/*f* noise of 4H-SiC SBDs increase gradually. Furthermore, the DLTS characterization clarifies that the Z_1/2_ defect is not the root cause of the device performance because the Z_1/2_ defect density is almost unchanged with the growth conditions. The AFM and PL tests further confirm that the crystal quality of the sample becomes worse with the increase of the C/Si ratio by comparing the RMS roughness and the SiC band edge emission intensity (391 nm). Therefore, it can be concluded that the increase of leakage current and noise are due to the crystal quality of 4H-SiC epilayers. The optimization of the C/Si ratio can significantly reduce TD density and improve the crystal quality of 4H-SiC epilayers, which further enhances the electrical properties of the 4H-SiC SBDs. Moreover, the study on the influence of C/Si ratio on TDs and Z_1/2_ defects should also be helpful for improving the responsivity of the 4H-SiC Schottky barrier UV photodetectors.

## Figures and Tables

**Figure 1 micromachines-11-00609-f001:**
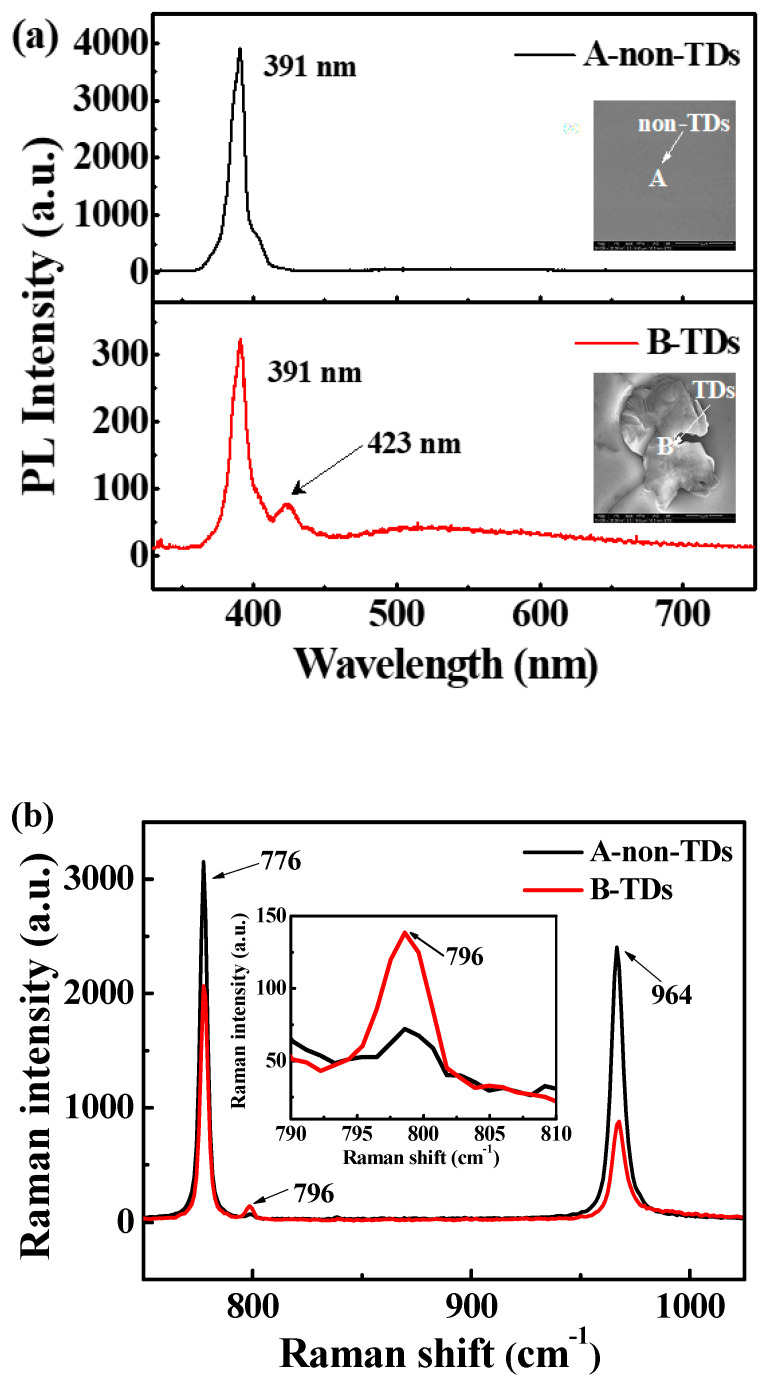
(**a**) Room temperature micro-photoluminescence (PL) spectra. The insets show scanning electron microscopy (SEM) images of region A (non-triangular defects (TDs)) and region B (TDs); (**b**) Micro-Raman spectra corresponding to the A and B positions in the 4H-SiC epilayer grown with C/Si = 1.1. The inset is a comparison of the intensity of the Raman peak at 796 cm^−1^ in the enlarged view.

**Figure 2 micromachines-11-00609-f002:**
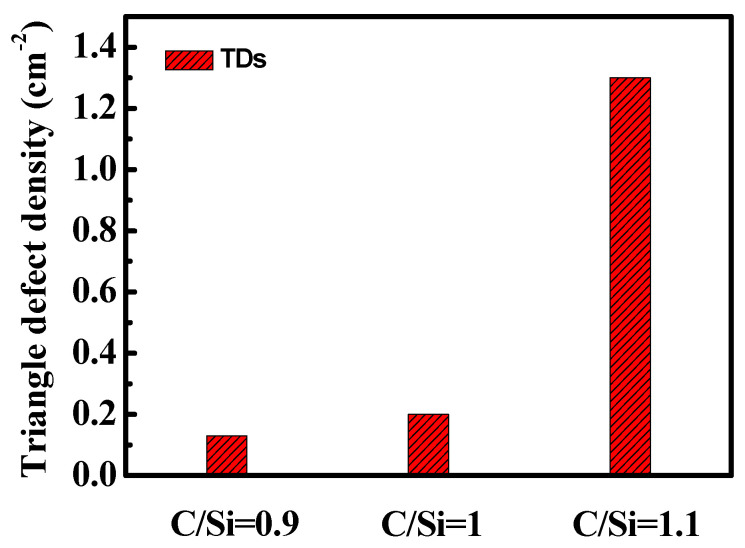
The C/Si ratio dependence of TDs density for 4H-SiC epilayers (#1~#3).

**Figure 3 micromachines-11-00609-f003:**
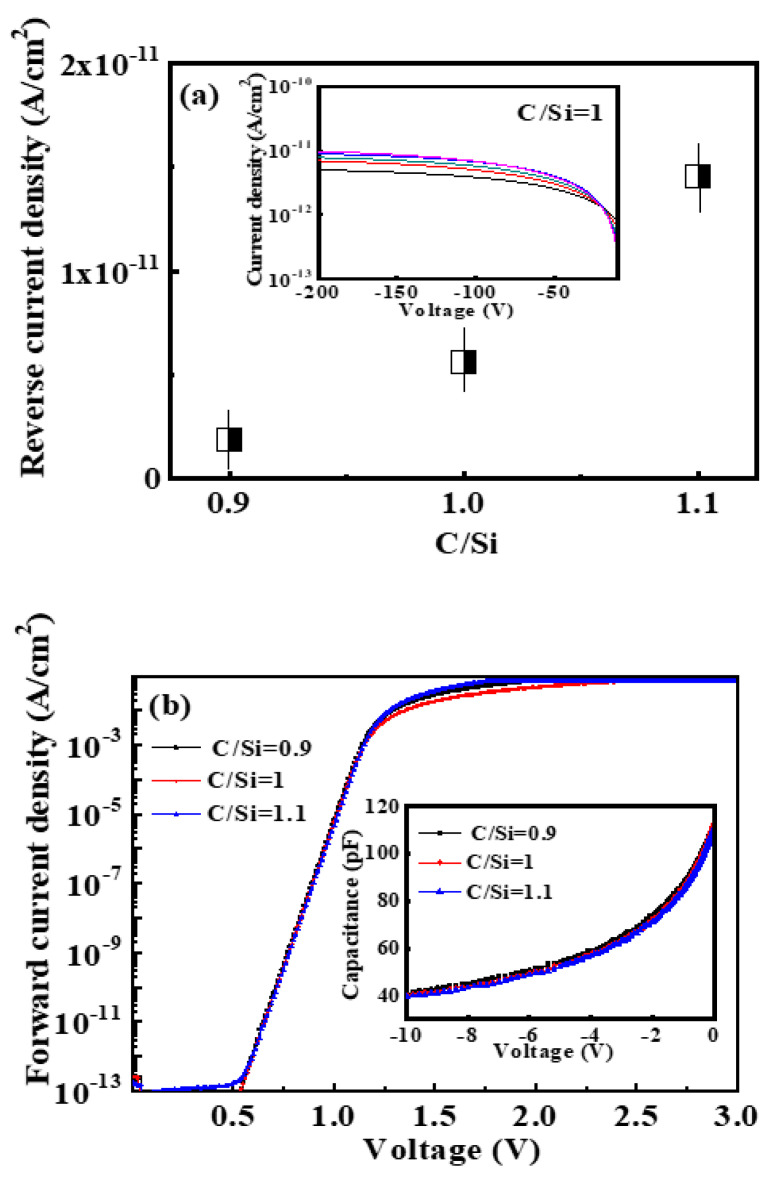
(**a**) The dependence of the mean reverse current density on C/Si ratio for Ni/4H-SiC Schottky barrier diodes (SBDs) under V_R_ = −200 V. The inset shows reverse *I-V* characteristics of a representative sample (Ni/4H-SiC SBDs with C/Si = 1); (**b**) Forward *I-V* characteristics of Ni/4H-SiC SBDs under different C/Si ratios (C/Si = 0.9, 1 or 1.1). The inset shows the *C*-*V* characteristics of Ni/4H-SiC SBDs; (**c**) Frequency dependence of the spectral noise density (S_I_) for Ni/4H-SiC SBDs with C/Si = 0.9 at room temperature under *V_R_* = −10~−200 V, the inset shows the bias voltage dependence of the spectral noise density at 1K Hz; (**d**) Noise spectra of Ni/4H-SiC SBDs with C/Si ratios (C/Si = 0.9, 1 or 1.1).

**Figure 4 micromachines-11-00609-f004:**
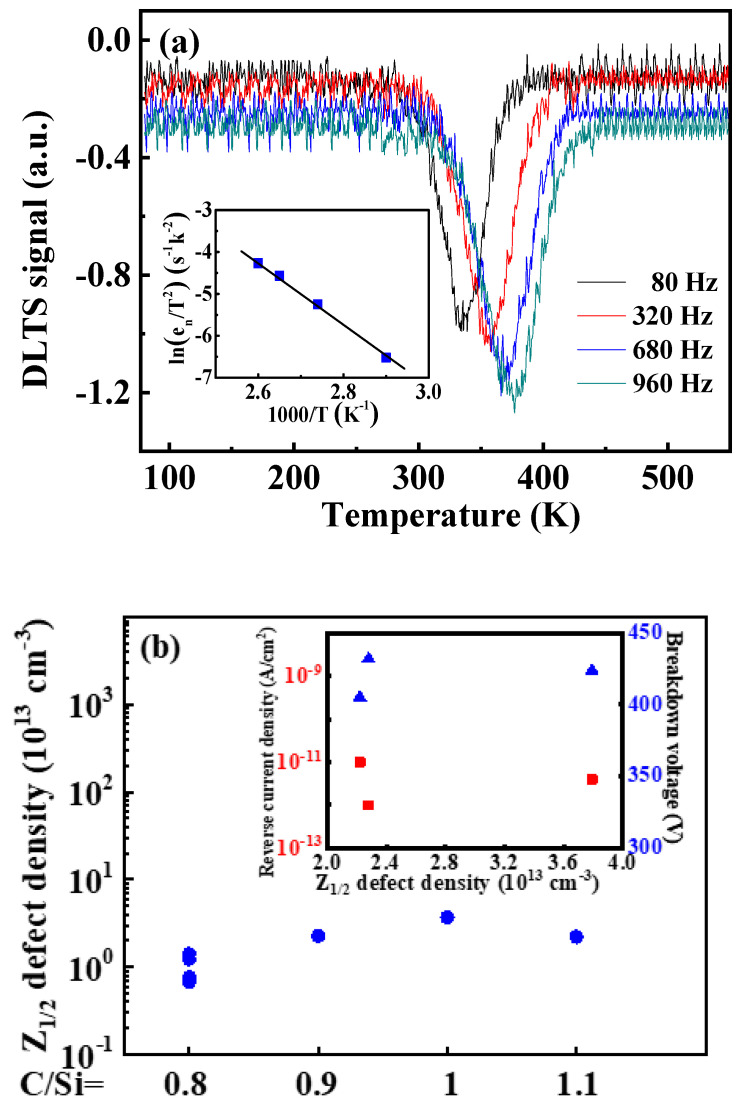
(**a**) Deep level transient spectrum testing (DLTS) spectra of Ni/4H-SiC SBD with C/Si = 0.9; (**b**) The Z_1/2_ defect concentration of Ni/4H-SiC SBDs under different CVD growth conditions. The inset shows the dependence of the reverse current density under *V_R_* = −200 V (red square symbols) and breakdown voltage (blue triangle symbols) on Z_1/2_ defect concentration for Ni/4H-SiC SBDs with C/Si ratios (C/Si = 0.9, 1 or 1.1).

**Figure 5 micromachines-11-00609-f005:**
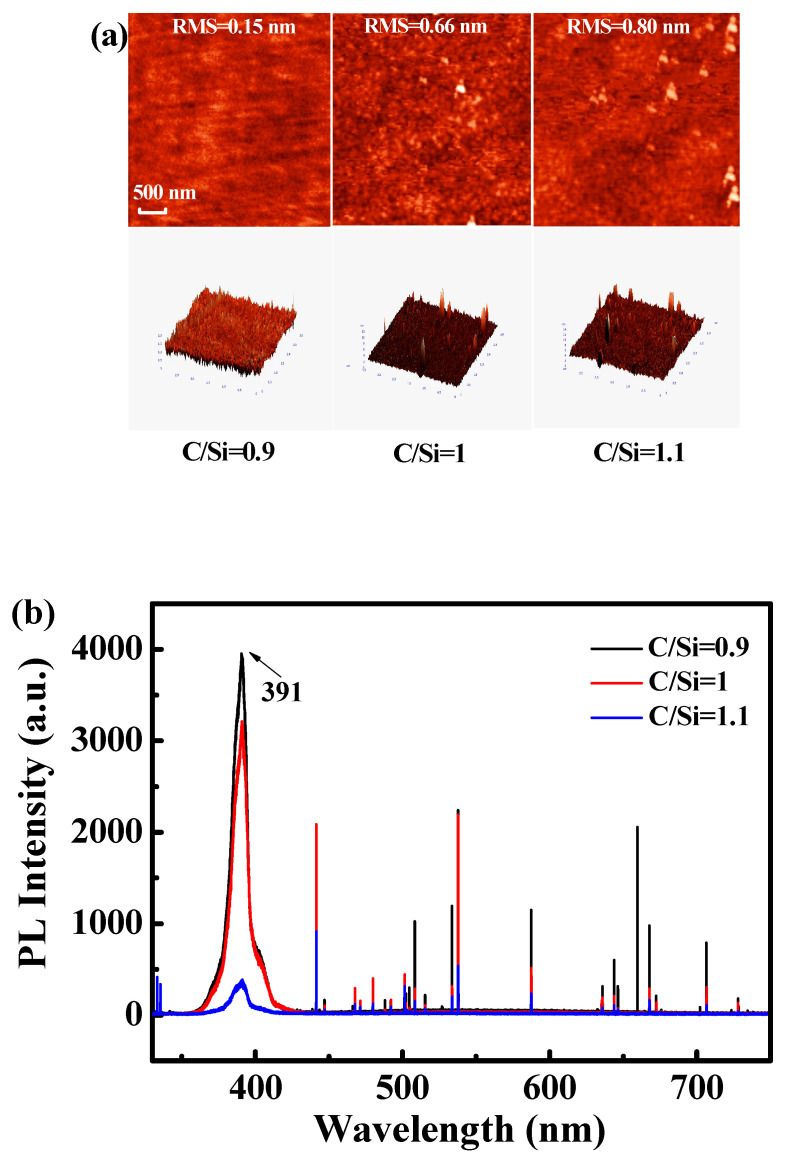
(**a**) Atomic force microscopy (AFM) images and (**b**) PL spectra of 4H-SiC epilayers with C/Si ratios (C/Si = 0.9, 1 or 1.1).

**Table 1 micromachines-11-00609-t001:** The samples of the 4-inch 4H-SiC epilayers with different chemical vapor deposition (CVD) growth parameters.

Samples #	#1	#2	#3	#4	#5	#6	#7
C/Si	0.9	1	1.1	0.8	0.8	0.8	0.8
Growth rate (μm/h)	60	60	60	60	60	60	30
Doping density (10^15^ cm^−3^)	1	1	1	4	7.5	10	7.5

**Table 2 micromachines-11-00609-t002:** The electrical parameters of 4H-SiC SBDs with different C/Si ratios (C/Si = 0.9, 1 or 1.1).

Samples #	#1	#2	#3
*n*	1.008	1.004	1.010
*Φ_B_* (eV)	1.629	1.631	1.629
*N_eff_* (10^15^ cm^−3^)	1.25	1.17	1.15

**Table 3 micromachines-11-00609-t003:** The Z_1/2_ defect parameters obtained from the DLTS measurements of the 4-inch 4H-SiC epilayers with different growth parameters.

Samples #	ΔE (eV)	σ (cm^2^)	N_t_ (cm^−3^)
#1	E_c_-0.627	1.18 × 10^−15^	2.28 × 10^13^
#2	E_c_-0.626	8.01 × 10^−16^	3.79 × 10^13^
#3	E_c_-0.624	5.06 × 10^−16^	2.22 × 10^13^
#4	E_c_-0.648	2.42 × 10^−16^	7.62 × 10^12^
#5	E_c_-0.644	2.44 × 10^−15^	6.78 × 10^12^
#6	E_c_-0.687	8.33 × 10^−15^	1.24 × 10^13^
#7	E_c_-0.610	5.96 × 10^−16^	1.41 × 10^13^

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
