# Peer review of "Effect of Various Defects on 4H-SiC Schottky Diode Performance and Its Relation to Epitaxial Growth Conditions"

_micromachines, 2020, doi:10.3390/mi11060609_

Round 1

Reviewer 1 Report

The manuscript reports on “Effect of various defects on 4H-SiC Schottky diode performance and its relation to epitaxial growth conditions” and is well written for the epi-layers quality with the defects as a function of the C/Si ratio. However, it should be addressed the following questions before the publication.

1. In Fig. 2 and Fig. 3, the density of TDs and the electrical parameters are improved as decreasing the C/Si ration from 1.1 to 0.9. According to this trend, the quality of the sample with 0.8 of C/Si ratio can be more improved. Please show the TD density and electrical properties of the epi-layer grown with 0.8 of C/Si ratio. 

2. For the inset of Fig. 4b, if the deep level is originated from an intrinsic point defect, it may affect to bulk electrical property such as ideality factor and forward and reverse breakdown voltages rather than the reverse current density, which may be related to an interface state. The authors should discuss on the breakdown voltages as a function of C/Si ratio or defect density.

Author Response

Author’s Responses to Reviewer’s Comments

Manuscript ID: micromachines-827746
Title: Effect of various defects on 4H-SiC Schottky diode performance and its relation to epitaxial growth conditions
Authors: Jinlan Li, Chenxu Meng, Le Yu, Yun Li, Feng Yan, Ping Han *, Xiaoli Ji *

We would like to thank the editors and reviewers for the constructive comments. Those comments are all valuable and very helpful for revising and improving our paper, as well as the important guiding significance to our researches. We have completed the revision following the instructions and comments of reviewers. We hope that the revised manuscript is now suitable for publication. Revised portion is highlighted in the revised manuscript. The main correction in the paper and the response to the reviewers’ comments are shown in the following attachment

Thank you for your consideration.

Sincerely yours,

Jinlan Li

Reviewer 2 Report

The manuscript presents results on CVD growth and the following characterization of 4H-SiC epilayers. The results obtained demonstrate the scientific reliability. The manuscript fits well with the aims of the journal but needs some revisions before it can be considered acceptable for publication.

(1) The paper should be re-organized in some way. Although there are 7 samples, only 3 of them are discussed. For better understanding, it will be useful to explain the aim of experiment with the C/Si ratio=0.8 and the choice of growth parameters in these experiments. Why doping density differs from #1-#3. Why you change the growth rate in the case of #7. And so on. Give more detail comments on #4-#7 and compare obtained results with #1-3.

(2) Z1/2 – please define this term in the text.

(3) A brief comparison of the results obtained with the literature is also necessary.

Author Response

(The authors gave the same response as above.)

Reviewer 3 Report

In this manuscript, the authors study effect of various defects on 4H-SiC Schottky diode performance and its relation to epitaxial growth conditions. They found at C/Si ratio of 0.9, resulting in a decrease of several orders of magnitudes in the noise level. Therefore, I recommend this manuscript for publication after revision. The authors should address the following:

  • In fact, for n-type semiconductors needs low work function metals for ohmic contact, but, author took Ni-metal as ohmic contact, could you explain.
  • The authors should add the C-V characteristics data.
  • In the noise spectra (Figure 3d), C/Si=0.9 follows 1/f behavior, but, why C/Si=1 and 1.1 follows 1/f2 behavior at higher frequencies.

Author Response

(The authors gave the same response as above.)

Round 2

Reviewer 1 Report

The authors clearly considered all comments of Referee. The results are interesting to the researchers in this field and are scientifically sound. Therefore, I recommend publishing this paper in Micromachines.

Reviewer 2 Report

I think that the authors have  improved the manuscript and now it can be accepted in the present form.